# Time-Specific Factors Influencing the Development of Asthma in Children

**DOI:** 10.3390/biomedicines10040758

**Published:** 2022-03-24

**Authors:** Daniele Russo, Mauro Lizzi, Paola Di Filippo, Sabrina Di Pillo, Francesco Chiarelli, Marina Attanasi

**Affiliations:** 1Department of Pediatrics, University of Chieti, 66100 Chieti, Italy; danielerusso1607@gmail.com (D.R.); mauro.lizzi.med@gmail.com (M.L.); difilippopaola@libero.it (P.D.F.); sabrinadipillo@gmail.com (S.D.P.); chiarelli@unich.it (F.C.); 2Pediatric Allergy and Respiratory Unit, Department of Pediatrics, University of Chieti, 66100 Chieti, Italy

**Keywords:** asthma, neonatal immune system, maternal smoking exposure, infections, children, gender, pollutants, adolescence

## Abstract

Susceptibility to asthma is complex and heterogeneous, as it involves both genetic and environmental insults (pre- and post-birth) acting in a critical window of development in early life. According to the Developmental Origins of Health and Disease, several factors, both harmful and protective, such as nutrition, diseases, drugs, microbiome, and stressors, interact with genotypic variation to change the capacity of the organism to successfully adapt and grow in later life. In this review, we aim to provide the latest evidence about predictive risk and protective factors for developing asthma in different stages of life, from the fetal period to adolescence, in order to develop strategic preventive and therapeutic interventions to predict and improve health later in life. Our study shows that for some risk factors, such as exposure to cigarette smoke, environmental pollutants, and family history of asthma, the evidence in favor of a strong association of those factors with the development of asthma is solid and widely shared. Similarly, the clear benefits of some protective factors were shown, providing new insights into primary prevention. On the contrary, further longitudinal studies are required, as some points in the literature remain controversial and a source of debate.

## 1. Introduction

Asthma is the most common chronic disease among children [1], exerting a substantial global burden that has increased greatly over the two last decades [2]. Globally, 14% of children are affected by asthma, with a death rate of approximately of 0.7 per 100,000 [2]. There are striking global variations in the prevalence of asthma symptoms in children, with up to 13-fold differences between countries [2]. Asthma is a complex and multifactorial chronic airway disease, clinically characterized by wheezing, shortness of breath, and cough, which can be caused by different exposures to allergens or infections [3]. Diagnosis is based on clinical symptoms associated with variable expiratory airflow limitation [3]. The presence or absence of respiratory signs show a poor inter-observer reliability, and the physical examination of people with asthma is often normal [4]. As a consequence, the diagnosis of asthma has to be supported by spirometry that seeks to demonstrate variable airflow obstruction [3]. However, spirometry influences the probability of a diagnosis of asthma due to its predictive value, which is characterized by a high positive predicting value, but a low negative predicting value [5].

Both genetic susceptibility and environmental exposures have been shown to be involved in the pathogenesis of asthma. There is increasing evidence that asthma has its origins in early life [6,7]. The Developmental Origins of Health and Disease (DOHaD) is a concept that arose over 30 years ago from the hypothesis of Professor David Barker, who examined how environmental factors such as nutrition, diseases, drugs, microbiome, and stressors interact with genotypic variation during the intrauterine phase of developmental plasticity to change the capacity of the organism to successfully adapt and grow in later life [8,9]. This could explain the complex inception of an individual’s risk of non-communicable diseases such as asthma in children and adults, giving the opportunity to develop therapeutic interventions during pregnancy and early life for the improvement of adult health [10]. The “fetal origins of asthma hypothesis” proposes that fetal adaptations to intrauterine and maternal conditions during development shape the structure and function of organs [11]; these alterations seem to impair lung function in childhood, subsequently resulting in wheezing and an earlier diagnosis of asthma [12]. Human lung development starts at approximately 4 weeks of gestation and includes four distinct histologic stages [13]. In addition, recent evidence has shown that a certain amount of alveolarization can occur up to early adulthood [14]. “Lung development after birth” would explain the reason why genetic and environmental insults occurring even after birth could induce long-term structural and functional impairment of the lung, predisposing individuals to the development of respiratory diseases in later life [15].

The aim of this review is to investigate the main predictive risk and protective factors for developing asthma in different stages of life: The fetal and perinatal period, infancy, childhood, and adolescence. These findings may help to understand the origins of non-communicable pathologies such as asthma and may contribute to the development of future preventive and therapeutic strategies for the improvement of health in later life. All the investigated predictive factors, classified in specific different stages of life, are illustrated in Figure 1.

## 2. Materials and Methods

We searched for articles on Pubmed using the keywords “asthma”, “neonatal immune system”, “maternal smoking exposure”, “infections”, “children”, “gender”, “pollutants”, and “adolescence”, preferring articles published in the last ten years. In particular, we used the following terms and logic: “asthma onset and fetal exposure or smoking and neonatal immune system or childhood lung function and infections or asthma prevention and vitamin d or prenatal life or childhood or adolescence”. Further studies were obtained through the references of some papers, in particular, meta-analyses with solid evidence published even before the time interval considered. Articles were selected according to their abstract, using eligibility criteria. The inclusion criteria were being in the English language, pediatric study population (age range 0–18 years old), and type of study: narrative and systematic reviews, retrospective analysis and prospective longitudinal birth cohort-studies, cross-sectional studies, and randomized control trials. Case reports, expert opinions, and manuscripts published in a language other than English were excluded. The final reference list was developed on the basis of originality and relevance to the broader scope of this review.

## 3. Fetal and Perinatal Exposure

Susceptibility to asthma is heterogeneous, as it involves both genetic and environmental insults (both pre- and post-birth), acting in a critical window of development in early life [16,17]. Studies suggest that in prenatal life there is an intrinsic immune fetal setting towards a T2 response, which is interestingly obliged to avoid possible rejective mechanisms of the fetus in the uterine environment [18,19]. The imbalance between the T helper 1/T helper 2 (Th1/Th2) response due to prenatal and postnatal environmental factors could deviate the fetal and neonatal immune system to enhance its response or divert it towards an abnormal activation to harmless stimuli, as we see in allergic diseases [18,19,20,21,22,23,24].

According to recent evidence, some conditions such as genetic factors, maternal infections and drugs, birth characteristics, and environmental exposures give information about the complex mechanisms shaping the immune and pulmonary system of neonates and infants, and the later possible development of asthma in children and adult life [21]. It therefore appears evident that current scientific research has invested resources in identifying early modifiable factors, which could allow the implementation of preventive strategies in order to change the decline of lung function in a reversible phase.

### 3.1. Genetic Factors

The heritability of asthma has been estimated to be around 60% [25,26], and large-scale genome wide association studies (GWAS) have identified multiple susceptibility loci [27,28]. Recent evidence suggests that single nucleotide polymorphisms (SNPs), such as in the interleukin 1 receptor-like 1 (IL1RL1) or on the chromosome 17q21, may play an import role in the risk and time-to-onset of asthma [26,27,28,29]. Although these results are promising to better understand the role of genetics, more studies are needed to confirm their effects in the development of asthma.

Another important element contributing to the risk of asthma in children is family history. In fact, it has been shown that maternal atopy and allergic comorbidities are associated with a higher risk of developing asthma in children [30]. Importantly, maternal asthma is the main risk factor associated with the development of early-life asthma, because it seems to impair the natural development of airways of children. These pathogenetic and still little-known pathways may lead to airway hyperreactivity and inflammatory cell mechanisms in children’s lungs [26]. In a meta-analysis including 33 studies, Lim et al. [31] showed that children born from mothers with asthma had approximately threefold greater risk of asthma than those born from mothers without asthma, especially if not well controlled.

### 3.2. Maternal Infections and Drugs in Pregnancy

Several maternal factors contribute to the risk of childhood asthma, such as respiratory and urinary infections, chorioamnionitis, and other maternal infections [26,32]. However, the exact pathogenetic mechanisms by which these factors determine a higher vulnerability of lungs to develop asthma later in life are still controversial. To the best of our knowledge, it seems that uterine exposure to microbial products, as well as to antibiotics, may play a key role in the onset of asthma and possibly in its specific phenotype. These results seem to derive from the intrauterine shaping of the lung microbiome and of the neonatal innate immune system [26,33].

To date, the effects of paracetamol are still controversial, which is why more studies are needed to better investigate its role in the onset of childhood asthma.

### 3.3. Allergen Exposure

In addition, several studies have investigated prenatal and postnatal early life exposure to common allergens [34], mold or/and dampness [35,36], and furry pets’ ownership [34,37], with consequent evidence of an increased risk of development of asthma and allergy in childhood [35,38,39,40].

### 3.4. Maternal Smoking Exposure and Oxidative Stress

To the best of our knowledge, it has increasingly been recognized that particulate matter (PM_2.5_ or PM_10_) and maternal smoking exposures (MSE) during the intrauterine period affect wheezing and childhood asthma susceptibility [41,42,43,44,45] and cause respiratory infections [46] and lung function impairment later in life [47,48]. Secondhand tobacco smoke during pregnancy could also influence the risk of asthma [49]. Prenatal PM_10_ exposure is likely to induce superoxide dismutase 2 (SOD2) protomer methylation in cord blood cells [50], which is related to phthalate and diisocyanate-induced asthma [51,52].

In particular, MSE is the largest modifiable risk factor for the development of asthma. Although the harmful effects of smoking are well-known, mothers usually cannot stop smoking due to nicotine addiction, even during pregnancy when nicotine metabolism is faster than when not pregnant [53]. A dose-dependent increase in asthma risk has been demonstrated in offspring due to MSE [53]. Early studies showed that cotinine, the stable metabolite of nicotine, could cross the blood-placental barrier and be found in fetal circulation and body fluids [54,55]. Thus, cotinine can bind the receptors in the lung to directly affect fetal lung development [54,55]. From a pathogenetic point of view, cigarette smoke, PM, and maternal obesity are notable environmental sources of inhaled free radicals and strong oxidants, leading to an increased oxidative stress in the maternal-placental-fetal unit [56,57]. The result is an imbalance between excessive oxidant activity and antioxidant capacity, because an excess of oxidants passes the blood-placental barrier. This harmful flow of oxidants determines increased reactive oxygen species (ROS), a reduction of endogenous antioxidant Manganese Superoxide Dismutase, and thus oxidative stress in offspring.

### 3.5. Intrauterine Growth Restriction

Oxidative stress determines various harmful effects on the immune and vascular systems of the developing fetus [58]. In fact, according to literature, oxidative stress seems to determine a higher vascular resistance in the uterine environment and, as a consequence, in the ascending aorta. This pathogenetic mechanism causes a damaging reduction of oxygen and nutrients to the developing fetus, leading to an impairment of the fetoplacental-maternal unit and intrauterine growth restriction (IUGR) [59]. IUGR is defined as the fetus’s inability to reach its potential growth, and it involves all neonatal systems, included the peripheral lung development. This impaired pathway could explain a future onset of asthma later in life [60,61]. Recently, den Dekker et al. [61] showed that childhood asthma and several other adult diseases could be influenced by different harmful variations of the natural fetal growth.

### 3.6. Environmental Exposures

Several environmental prenatal factors seem to influence the development of asthma and atopy in children, such as persistent and non-persistent organic pollutants (POPs and nPOPs), volatile organic compounds (VOCs), and dust [62,63]. These widespread substances contribute to airway inflammation and impair lung function performance in children, with a higher risk of developing asthma later in life [64,65,66,67,68,69,70].

In addition, studies suggested that prenatal exposures to metals such as mercury, arsenic, and cadmium could increase the risk of asthma in offspring, likely due to an impairment of the neonatal immune system, deviating it to a prevalent Th2-response and a decreased Treg suppressor function [21,71]. The results of this altered immune system may be, on one hand, a higher susceptibility to respiratory infections during the early postnatal period, and, on the other hand, an increased risk of wheezing and asthma in childhood [66,67].

### 3.7. Endocrine Disruptors

Endocrine disruption during pregnancy is a potential cause of adverse pregnancy outcomes. These results may be caused by one of the harmful effects of smoking, such as its anti-estrogenic mechanism in pregnant mothers, leading to abnormal prenatal hormonal patterns and, possibly, to impaired lung function later in life. Indeed, it has been shown that smoking mothers have reduced estrogen levels in the cord blood [72]. Moreover, endocrine-disrupting chemicals (EDCs) and polycyclic aromatic hydrocarbons (PAH) on the surface of PM seem to alter sex hormone synthesis [73,74].

Although these are promising results, further studies are needed to better investigate the exact correlation between prenatal exposure to endocrine disruptors and future risk of asthma in the offspring [75].

### 3.8. Epigenetics

Mounting evidence has widely shown how environmental factors contribute to the increased risk of childhood asthma, even due to their influence on epigenetic programing.

Indeed, MSE, PM exposure, pollutants, and toxicants seem to stimulate TH2 responses, resulting in mucin hypersecretion, eosinophils accumulation, and, eventually, to pulmonary diseases [76]. These pathogenetic results may derive from epigenetic mechanisms, which can be defined as inheritable molecular pathways linking genetics with environmental exposures. Different studies have investigated the role of harmful epigenetic changes, such as DNA methylation, histone modifications, and micro–RNAs (miRNAs) [77,78,79,80,81,82,83,84]. The results highlight how these alterations of DNA sequence explain not only the rise of complex diseases later in life, such as asthma and chronic obstructive pulmonary disease (COPD), but also their endophenotypes and specific characteristics [85,86,87,88].

The investigation of such risk factors acting in a critical window of development in early life may help to understand and predict the later possible developing of asthma in children and adult life.

### 3.9. Prenatal Farming Exposure

Potential prenatal and perinatal factors have been studied as protective elements for the risk of developing asthma in later life.

Recent evidence has shown that a farming environment and ownership of pets could reduce the risk of asthma, allergic diseases, and atopy [89]. These results may derive from the prenatal environmental shaping of the innate immune system, with a reduction of Th2 response (IL-5) and a powerful Th1-related (IFN-γ) pathway, which have been observed in neonates born from parents living in farming environment [21,90,91,92]. In addition, these neonates, especially those exposed to dogs and endotoxins, seem to have decreased total IgE levels and specific IgE to seasonal allergens in cord blood, compared to those exposed to cats and dust mite.

Therefore, these findings may help to understand some possible protective mechanisms preventing the onset of complex and widespread diseases in childhood such as asthma.

### 3.10. Vitamins

McEvoy et al. [93] investigated the role of supplementation of vitamin C in smoking women during pregnancy, highlighting the capacity of this molecule to improve lung function (better airflow and less wheezing) in children during the first year of life. Vitamin C, indeed, is an antioxidant contributing to cellular antioxidant defense [94]; in this sense, vitamin C and other antioxidant supplementation during pregnancy, such as L-carnitine, could completely or partially reverse the adverse effects on those organs induced by MSE [95,96].

Interestingly, two meta-analyses (the first one including 32 studies and the second one 11 studies), observed that women with vitamin D levels of ≥30 ng/mL at randomization and who assumed a high intake of vitamin D and/or fish oil during pregnancy showed almost halved risk to have children with wheeze in infancy [97,98,99].

Moreover, Bisgaard et al. [100] conducted a double-blind, placebo-controlled, parallel-group trial, including 736 pregnant women between 22 and 26 weeks of gestation, randomly assigned to receive n-3 long-chain polyunsaturated fatty acids (LCPUFAs, fish oil) or placebo. The results of this study showed a great reduction of risk of persistent wheezing or asthma by approximately 7 percentage points, or one third, in the first 5 years of life among children of women undergoing daily supplementation with n-3 LCPUFA during the third trimester of pregnancy.

In a birth cohort of 1924 subjects, Turner et al. [101] suggested a direct link between vitamin E status, fetal size, and pulmonary function. Indeed, vitamin E seems to have a crucial role in fetal development and, in particular, in fetal lungs. Therefore, mothers with low levels of vitamin E (a-tocopherol) showed a reduction of fetal size in the first trimester, resulting in compromised neonatal and childhood respiratory function and, eventually, in higher risk of developing asthma. In this sense, specific nutritional interventions may help to prevent a future risk of asthma later in life.

### 3.11. Prematurity

In literature, several retrospective studies and meta-analyses associated prematurity with an increased risk of respiratory symptoms and asthma in children [102,103,104,105]. The immature lungs of preterm newborns are predisposed to a suboptimal lung development [106].

A premature birth before 32 weeks of gestation was associated with an increased risk of wheezing in preschool children and at 7 and 11 years of age [107,108]. Additionally, several studies showed that rapid weight gain in preterm infants or low birth weight are independent risk factors for adverse respiratory outcomes [61,109]. A meta-analysis of 31 European cohort studies with 147,252 children up to the age of 10 years confirmed the association of shorter gestational duration with the risk of asthma development [110]. Preterm-born children had an increased risk of preschool wheezing and school-age asthma compared to term-born children, independently of birth weight.

In literature, it is not established the exact length of gestation required to reduce the asthma risk. A large Swedish study with 622,616 subjects showed that only extremely preterm birth (<28 weeks of gestation) was associated with an increased risk of asthma in young adulthood (25.5 to 35 years of age) [111]. In a Swedish national cohort of 1,100,826 children, Vogt et al. [112] observed that the decline of asthma risk was positively associated with increasing gestational duration at 6 to 19 years of age, even in children born at term. Other factors related to prematurity, as the use of mechanical ventilation, oxygen supplementation and bronchopulmonary dysplasia, contribute to an increased risk of asthma [113,114]. Bronchoalveolar structural immaturity due to premature birth in such a critical stage of fetal development, or the lung damage due to neonatal treatment, are the main causes in prematurity of chronic lung disease [115,116].

The duration of asthma risk related to prematurity is not still clear. The effect of prematurity on respiratory symptoms was stronger in the first years of life [102,112] and decreased progressively in adulthood [111]. Noteworthily, adverse early life events affect lung growth, compromising the achievement of “personal best lung function” and predisposing individuals to chronic obstructive pulmonary disease in adulthood [117]. Nowadays, studies on respiratory outcomes related to preterm birth often produce contrasting results because of several methodological problems, such as the heterogeneity of populations, definitions, treatments, birth characteristics, and environmental exposures [117,118].

To conclude, long-term studies with large and homogenous populations are needed to better investigate the effect of prematurity on lung function and asthma development across life phases.

### 3.12. Birth Characteristics

It is widely shown that the children born in cold seasons have a higher risk of asthma compared with those born in spring, likely due to higher lymphocyte and leukocyte counts and concentrations of pro-inflammatory cytokines [21].

Mode of delivery appears to influence the occurrence of asthma and allergy in later life. Cesarean delivery (CD) has greatly increased in the last two decades [119], with a higher risk of childhood asthma as well [120,121]. Several studies found that CD could determine different mechanisms, such as an enhancement of Th2 response and an increased secretion of IL-13 to allergens, phytohemagglutinin (PHA), and lipopolysaccharide (LPS), leading to a rising risk of recurrent wheezing [120,122]. In addition, a meta-analysis of 13 studies including 887,960 participants showed that children born from CD had a higher risk of asthma up to the age of 12 [123]. Another meta-analysis recently conducted by Wypych-Ślusarska et al. [124] including 41 articles also investigated the relationship between delivery by Caesarean section and asthma in children. The results were controversial due to the significant heterogeneity of the studies; thus, more information is needed to better clarify this connection.

The risk of immune diseases, including asthma, and the severity of respiratory infections [125,126,127] due to cesarean delivery likely derive from the absence of stress factors and immune-dysregulation, which are typically generated from vaginal delivery. This impaired natural course determines an altered gut microbiota immune programming in the offspring at birth [128]. In this sense, the stress of labor can be associated with the activation of specific immune cells and an adaptive switch towards a Th1 response, reducing the risk of developing asthma in adolescents and adults [129]. Furthermore, CD influences the variety of microbial agents that colonize the gut: newborns are exposed to the maternal skin flora instead of birth canal microbes [130].

The composition of the microbiota is strongly influenced by mode of birth, as aforementioned, but also by host genetics and immunity, diet, infections, antimicrobial agents, and family composition [131]. Children born by vaginal delivery showed an intestinal colonization with predominance of *Lactobacilli* and a higher flora/microbiological diversity, compared to gut dysbiosis in children born by Cesarean section [132].

It is hypothesized that the higher prevalence of chronic inflammatory diseases in Western countries may be correlated with permanent loss of bacteria in the resident microbiome. Several authors have shown that children with asthma had a reduced variety of intestinal bacteria at one month of life, compared to subjects without asthma [133,134]. A recent study observed an association between the metabolic profile of children born through CD and an increased risk of asthma in school age, assuming a connection with early life gut microbial dysbiosis [135]. These findings support the association between CD and the risk of childhood asthma through a perturbed immune responses and gut microbial colonization patterns revealed in the blood metabolome at birth. The concept that microbial-derived metabolites or immunomodulatory molecules produced in the gut could transit towards the lung to exert biological activities is known as the gut-lung axis [136]. However, data are still partial and conflicting; rigorous metabolomic studies could better clarify this aspect in the future.

In addition, greater risk for obesity and delayed breastfeeding are other factors related to CD that could represent an additional indirect effect of CD on asthma development. Some studies found a higher risk of obesity in children born by CD [137,138]. In a prospective cohort study with 22,068 offspring of the Growing-Up-Today-Study, the adjusted risk ratio for obesity in offspring delivered via Cesarean birth was 1.15 compared to those delivered via vaginal birth [137]. Since obesity is thought to be another risk factor for asthma [139], this may be an additional indirect effect of CD.

### 3.13. Breastfeeding

Breastfeeding is often delayed in CD infants [140], hampering the physiological establishment of the microbiome [141]. However, a meta-analysis of 23 studies showed that the association between CD and asthma was independent of confounding factors such as breastfeeding duration, maternal smoking, and low birthweight [142].

Human milk is the optimal source of nutrition for term infants during the first six months of life [143]. Breast milk is an immunologically living biologic substance, containing multiple compounds that contribute to infant growth and promote development of host defense mechanisms [144]. To date, studies about the association of breastfeeding with asthma are limited and controversial because randomizing subjects to breastfeeding or formula feeding is unethical. Moreover, heterogeneity in asthma and breastfeeding assessment in study settings and human milk composition are potential source of bias across the different observational research studies [145]. Lodge et al. [146] found that a longer breastfeeding duration was associated with a reduced risk of asthma between 5 and 18 years of age. In a meta-analysis of 117 studies, Dogaru et al. [147] showed that the protective effect of breastfeeding on asthma was greater during the first 2 years of life, when wheezing is often associated with airway infections and not necessarily with the later development of asthma [148]. The greater effect in this age group could be explained by the immunological effect of breastfeeding in reducing airways infections and therefore, episodes of wheezing [149]. The persistence, albeit diminished, of protection in school-aged children is consistent with the hypothesis that the development of later asthma could be triggered by respiratory infections in early life [150]. A systematic review and meta-analysis suggested that the duration and exclusivity of breastfeeding are associated with a lower risk of asthma, but there was no statistically significant effect on the ≥7-years age group [151]. As the child grows, the complex interaction between the various predisposing factors becomes evident. A recent study showed that statistical significance by the age of 6 years was reached only in children without a family predisposition to asthma [152].

### 3.14. Neonatal Jaundice

A recent meta-analysis showed that children who had jaundice and received phototherapy during the neonatal period were more likely to develop childhood-onset asthma [153]. The pathogenesis of this association remains unknown. It has been hypothesized that the cumulative effect of unconjugated bilirubin and DNA damage secondary to phototherapy may induce the Th1/Th2 switch disorder [154]. The subsequent Th2 predominance may in turn contribute to the development of allergic diseases [155]. However, the association between neonatal jaundice and childhood asthma is probably smaller than previously estimated.

## 4. Infant Exposure

### 4.1. Antibiotics and Paracetamol

Ren et al. [156] in a recent cross-sectional study including 6183 preschool children (3165 boys and 3018 girls), investigated several infant exposures contributing to a higher risk of developing asthma. In particular, this study showed that some relevant factors involving antibiotic use and paracetamol in the first year of life were importantly related to asthma.

The reason for the strong connection between antibiotics and asthma may be the consequent alteration of the natural gut microbial system and impaired immune system, especially in the first years of life. The result of such modifications could contribute to a higher susceptibility to developing asthma [157].

In addition, Gonzales-Barcala et al. [158] conducted a cross-sectional study on more than 20,000 children and adolescents about the effects of paracetamol on asthma. The results obtained provided evidence about a connection between paracetamol use, likely in a dose-response manner, and an increase in asthma prevalence, although the effect of paracetamol on asthma onset are still matter of debate.

### 4.2. Vitamin D

Vitamin D is a modulator of the immune system and is involved in regulating cell proliferation and differentiation. Recent evidence suggested that vitamin D deficiency may predispose one to allergic diseases [159,160,161]. In vitro studies showed that vitamin D suppresses IgE production by human B lymphocytes, increases Interleukin 10 (IL-10) production with the promotion of a regulatory B-lymphocyte phenotype, and suppresses mast cells activation [162]. A vicious circle of causality and reverse causality between vitamin D and asthma onset makes difficult to define their relationship. Findings from epidemiologic studies suggest a significant association between reduced levels of serum vitamin D and an increased risk for asthma and reduced lung function [163]. In addition, vitamin D participates in antimicrobial immune responses [164,165]. A meta-analysis with 10,933 participants from 25 randomized control trials showed that vitamin D supplementation reduced acute airways infections in patients with a severe vitamin D deficiency [166], removing a possible trigger for the development of asthma in later childhood. Similarly, a literature review of randomized control trials found that vitamin D supplementation in asthmatic children older than 2 years led to a significant reduction in asthma exacerbations [167]. Contrarily, a recent randomized, double-blind, placebo-controlled clinical trial (The Vitamin D to Prevent Severe Asthma Exacerbations [VDKA] Study), showed that vitamin D3 supplementation did not significantly improve the time to a severe exacerbation in 192 asthmatic children with serum vitamin D levels less than 30 ng/mL [168]. To date, therefore, no clear evidence exists regarding the supplementation of vitamin D in childhood asthma primary prevention [169,170,171].

### 4.3. Viral Infections

The first years of life represent a crucial period for alveolarization, establishment of microbiomes, and immune system development [172,173,174]. Simultaneously, the greatest peak in the incidence of lower airways infections occurs during this age [175]. Respiratory syncytial virus (RSV) and human rhinovirus (HRV) are the most common etiologies of acute airways infection in infants and preschool children [176]. These viral agents are predictive of the development of asthma and impaired lung function later in childhood [177,178,179], although there is still disagreement about a causal effect. In a recent systematic review of 41 studies [180], Shi et al. found that infants who contracted RSV had an odds ratio of 3.05 for recurrent wheezing up to 36 months of age, and an odds ratio of 2.95 for asthma up to 12 years of age. Lu et al. [181] showed how a more severe RSV-related bronchiolitis correlated with greater likelihood for asthma later in childhood. This finding suggested that an exaggerated inflammatory response to RSV could reflect inherent bronchial hyper-responsiveness. A six-year follow-up of the MAKI trial investigated 395 high-risk preterm infants who had randomly received either palivizumab or placebo during the RSV season of their first year of life [182]. The authors found that RSV prevention did not affect the risk of physician-diagnosed asthma or the lung function tests in school aged children. HRV infections are more frequent after 1 year of age [183].

In a cohort of 217 children followed prospectively from birth to 13 years of age, an additive effect of early life aeroallergen sensitization and HRV-related wheezing on asthma risk at adolescence was found [184].

In addition, Magnier et al. [185], conducted a study including 222 infants, prospectively followed to age seven. The study investigated the link between children who had a first episode of acute bronchiolitis due to HRV with a family history of atopy and asthma, highlighting the possibility of a higher risk of developing asthma at school age.

In a Finnish longitudinal study with 408 infants, the authors recognized three bronchiolitis profiles: Profile A, characterized by history of recurrent wheezing and/or eczema, wheezing during acute disease, and HRV infection; Profile BC, characterized by severe illness and RSV infection; and Profile D, which includes subjects with milder disease, mostly without wheezing and with HRV infection. The highest risk of childhood asthma was found in children with Profile A [186]. These latter findings, together with the recent report of interaction between early airways infections and the asthma locus at chromosome 17q21 [28] or another risk gene, *CDHR3* [187], suggest that HRV plays a key role. This pathogen could be a revealing factor for those with early airway inflammation or an early marker of impaired antiviral response.

To date, asthma is considered an umbrella diagnosis for several diseases with distinct underlying molecular pathways (endotypes) [188]. The subgroups are secondary to many genes interacting with various environmental exposures, and these interactions are not unequivocal [189].

Lemanske et al. [190] in their COAST (Childhood Origins of Asthma) study, including a cohort of 287 children, sought to investigate the complex asthma inception studying the role of age, cytokine production, and virus infections. They observed that the interactions among these three factors, acting in a critical window of development of immune system or lung, contribute to explain the future risk of asthma inception in the first decade of life.

It is difficult to interpret and predict the effect that a specific exposure may have on each subtype. However, a better knowledge of the underlying mechanisms could allow us to understand, which categories would benefit from preventive strategies or tailored novel therapies.

### 4.4. Day-Care Attendance and Parental Socioeconominc Status

A systematic review of 18 studies [191] found that day-care attendance could be a risk factor for the development of early recurrent wheezing during the first three years of life and asthma before the age of six. Nevertheless, after this age it was no longer a risk factor. The limits of this data are represented by a high heterogeneity between the studies, probably due to observational design and variability in the measurement of exposure and effect.

Noteworthily, Caffrey Osvald et al. [192] conducted an interesting register-based cohort study, including 955,371 individuals, on a possible association between parental socioeconomic status (SES, measured as education/income) and asthma or wheezing in offspring. The results of this studies suggested how parental education could be linked to asthma onset and phenotypes, not only due to genetic or environmental exposures shared by first cousins, but also as a consequence of pathogenetic mechanisms not still fully understood. Therefore, further studies are needed to examine in depth causal relationships between these factors and to develop strategic interventions to predict and improve children’s health.

### 4.5. Atopy

Asthma phenotypes are strongly linked to early allergic sensitization [193]. The association of asthma with other atopic conditions (e.g., dermatitis and rhino-conjunctivitis) is well-documented in literature, although sensitized subjects do not necessarily develop allergic disease [194]. Asthma is more common in children sensitized to aeroallergens early in childhood than their non-sensitized peers. Similarly to asthma, recent evidence has shown that ‘atopic sensitization’ is heterogeneous with several distinct sub-groups [195]. A recent study by Malby Schoos et al. [196] showed that sensitization to multiple allergens and persistent sensitization during childhood were associated with increased risk of asthma at 13 years of age. Belgrave et al. [174] showed four distinct trajectories of lung function from preschool age to young adulthood, integrating data obtained in population-based birth cohort studies: persistently high, normal, below average, and persistently low. The authors found that persistently low FEV1 was associated with childhood asthma and tobacco smoke exposure, and was predicted by severe recurrent wheezing and early allergic sensitization. It is not known which allergen is most strongly associated with asthma development. However, the Manchester Asthma and Allergy Study (MAAS) showed that early sensitization to multiple aeroallergens was strongly associated to severe early onset asthma [197]. There is conflicting evidence that early sensitization has a causal link towards respiratory disease and impaired lung function in adulthood [198,199]. A deeper knowledge of the existing relationship could clarify which subgroup of patients could benefit from primary and secondary prevention measures. To date, studies investigating the impact of allergen immunotherapy on lung function have showed no effects on FEV_1_ and FVC, but small effects on MEF_25–75_ [200].

### 4.6. Infant Farming Exposure

The lower prevalence of asthma in populations highly exposed to microbial environments, such as traditional European or Amish farms, suggests a significant role of environmental microbiome in asthma development [201,202]. Data that emerged from a meta-analysis of birth cohort studies showed that infant farming exposure was significantly associated with 25% lower asthma prevalence [203]. The protective effect of farm life on childhood asthma may be secondary to exposure to various environmental agents in stables and farms, as well as to nutritional factors, particularly the consumption of raw cow’s milk [204]. To conclude, genetically-mediated alterations of immune response toward both respiratory viruses and aeroallergens, and the absence of protective mechanisms such as early microbial exposures, may predispose individuals to severe and persistent early onset asthma phenotypes.

### 4.7. Air Pollution

Growing evidence suggests that early-life exposure to indoor and outdoor air pollution caused by urbanization and population growth contributes to the development of asthma in children [205].

Outdoor air pollutants such as nitrogen dioxide (NO_2_), particulate matter with a diameter smaller than 2.5 μm or 10 μm (PM_2.5_ or PM_10_), or ozone (O_3_) can cause airway inflammation and hyper-responsiveness [206].

In addition, Muñoz X. et al. [207] have recently investigated the correlation between diesel exhaust particles (DEPs), the solid fraction of the complex mixture of diesel exhaust and asthma. DEPs may have two main different pathways involved in the rise of asthma: A direct way, with oxidative stress and the induction of a Th2 response; and an indirect mechanism, involving epigenetic alterations and microbiome changes.

Gauderman et al. [208] showed that long-term improvement in air quality was associated to statistically and clinically significant positive effects on lung function growth in children. Therefore, postnatal exposure to high levels of air pollution is associated with lower lung function and functional growth [208]. Indoor air pollution from the use of open fires for cooking increases the risk of developing asthma in children in low-income countries. Solid fuels or biomass are used as primary fuels for cooking or heating in many countries; reducing this energy source could represent an important risk control strategy in the future [209].

### 4.8. Gastroesophageal Reflux Disease

Cantarutti et al. [210] conducted a retrospective cohort study including 86,381 children to investigate the association of treated and untreated gastroesophageal reflux disease (GERD) in the first year of life and the risk of asthma after 3 years. The results showed that an increased rate of asthma onset was recorded in children with GERD compared to controls, despite any medications. The reasons behind such higher risk may be the chronic irritation of the airway tract exposed to gastric acidity or constant inflammation causing bronchospasms due to continuous nerve stimulation [211].

Further studies may help to better understand the underlying mechanisms of GERD and all the possible ways it can lead to the development of asthma later in life.

## 5. Adolescent Exposure

### 5.1. Obesity

Several studies stated that rapid weight gain after prematurity is an independent risk factor for adverse respiratory outcomes [61,212]. There is also evidence that body-mass index (BMI) is positively associated with asthma symptoms in childhood, especially in high-income countries [213]. A meta-analysis of six prospective cohort studies including 25,734 children showed that obesity (defined as BMI ≥ 95th percentile) was significantly associated with a 50% increased risk of asthma in both boys and girls, with no heterogeneity among studies [214]. Childhood obesity is associated with airway dysanapsis, an imbalance between growth of the lung parenchyma and airway caliber, reflected by normal or supranormal FEV_1_ and FVC, but with larger effects on FVC, leading to a low FEV_1_/FVC ratio [215]. Beyond mechanical changes, obesity has also immunomodulatory effects. The pro-inflammatory role of leptin and resistin supports the hypothesis that obesity may lead to new-onset asthma [216].

Yang-Ching Chen et al. [217], in a recent study involving 7069 children aged 12, investigated multiple obesity-related risk factors, such as obesity, physical activity, cardiopulmonary physical fitness, sleep-disorder breathing (SDB), and sleep quality, showing some possible explanations linking their role with childhood asthma. In particular, the aforementioned factors seem to enhance a Th2 inflammatory response and the productions of inflammatory marks such as IL-6 and NF-kB, which may worsen airway hypersensitivity.

To conclude, asthma and obesity might have additive synergistic proinflammatory effects, even though more studies are needed to provide preventive and therapeutic interventions.

### 5.2. Tobacco Smoke

Some studies have highlighted positive associations between postnatal environmental tobacco smoke (ETS) exposure and risk of recurrent wheeze and asthma in early childhood [218,219,220]. Nevertheless, cofounding factors make the strength of this association unclear and the results of the effect of passive exposure on lung functional growth are heterogeneous [218,221]. In their meta-analysis, Silvestri et al. [222] found a borderline association for postnatal exposure to parental smoking with asthma in childhood, with contradictory data for schoolchildren and adolescence due to the limited number of studies available. In part, these results may be due to the difficulty of evaluating postnatal exposure alone, independently of prenatal exposure. The association with a higher risk (OR = 1.23; 95% CI, 1.01–1.51) was instead found in a study by Thacher et al. [223], who noted that the influence of ETS exposure was strongest on asthma in preschool age. Similarly, He et al. [224] reported more recently a 24% increased risk of developing asthma. Early exposure to ETS may have a long-term effect on lung function, persisting until adulthood [84,225].

Hospital admission rates for asthma among pre-school and school-aged children were significantly reduced as an effect of the ban of smoking in public places [226,227]. Several longitudinal studies demonstrated the increased risk of asthma in adolescents exposed to active smoking [228,229]. Guerra and colleagues contributed to the hypothesis that parental smoking and active smoking act synergistically on the development of airflow limitation and accelerated lung function decline in young adults [230], contributing to the persistence of asthma in addition to its occurrence. In addition, recent studies found a significantly higher prevalence of asthma among adolescents who reported E-cigarette smoking [231,232]. The mechanism of possible e-cigarette effects on pulmonary function is not clear. E-cigarette vapor may produce inflammatory responses, increased airway resistance, and increased susceptibility to infection [233,234].

### 5.3. Gender and Puberty

A different gender prevalence of asthma is present and changes with age [235]. In prepubertal age, males exhibit symptoms of wheezing and/or asthma more frequently compared to females [236]. The gender disparity in asthma prevalence in children may be partly explained by allergen sensitization or different body mass index (BMI) [237,238,239]. Furthermore, differences in immune responses and anatomical peculiarities might provide some rationale for the gender difference in wheezing and/or asthma in boys and girls when sex hormones are low [240]. Interestingly, around puberty, clinical evidence shows that increased asthma symptoms occur in females compared to boys [241]. By adulthood, the prevalence of asthma is increased in women compared to men [242]. A role of sex hormones in the etiology of asthma was supposed, but the exact biologic mechanism remains unclear. Several studies showed an association between early menarche (<12 years old) and the risk of asthma [243,244,245], suggesting that estrogen exposure and absence of progesterone exposure could increase the risk [243]. Testosterone suppresses the type 2 response, whereas estrogen and progesterone enhance type 2 and suppress type 1 responses in females [246]. This may partly explain the higher prevalence of asthma in female gender after puberty.

### 5.4. Nutrition

Gonzalez Barcala et al. [247], in a cross-sectional study including 14,700 children and adolescents, investigated the role of nutrition and in particular, of Mediterranean diet (MD) in the onset of asthma. Although there were some promising findings in asthma pathogenetic pathways, such as a reduction of bronchial inflammation and an immunomodulation to a Th1 response, no protective effects have been observed on the prevalence of asthma.

Another relevant study evaluating the impact of nutrition on asthma was conducted by Cilluffo et al. [248], in a cross-sectional study on 415 children aged 5–14 years. This study investigated the connection between the Dietary Inflammatory Index (DII), a new marker of dietary inflammation, and asthma outcomes. Two classes of asthmatic children were found, with different asthma manifestations according to different indoor exposures. It seems that proinflammatory dietary patterns result in higher impact of symptoms in children and adolescents with persistent asthma. Although these are promising results, more studies are necessary to clarify whether different dietary patterns could change and improve asthma outcomes.

## 6. Conclusions

Proposed risk factors for asthma vary with the age of asthma onset and timing of exposure. Adverse exposures during fetal and postnatal life could impact lung growth and development, with consequences of persistently smaller airways and compromised lung function [6]. Various pathways, including adverse infant exposures and the consequent growth adaptations and respiratory health outcomes, have been studied. After reviewing the literature, we conclude that significant scientific evidence supports the hypothesis of fetal origins of the lung diseases and that even exposures after the first years of life could influence the development of asthma later in life. However, the mechanisms by which specific adverse factors in early postnatal life lead to respiratory disease in adult life are still not fully understood. More studies are needed to better understand the multiple pathogenetic pathways and exposures acting in the complex and heterogeneous inception of asthma. These findings may help to translate these exposures into prevention strategies and therapeutic interventions to predict and improve health later in life.

## Figures and Tables

**Figure 1 biomedicines-10-00758-f001:**
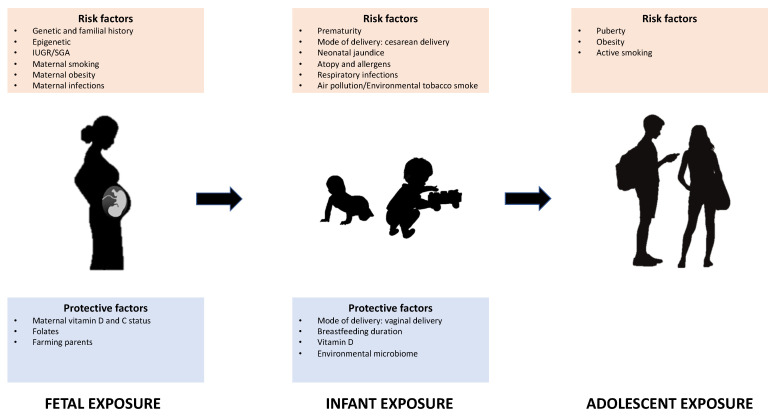
Main risk and protective factors for the onset of asthma in different stages of life. IUGR: intrauterine growth restriction; SGA: small for gestational age.

## Data Availability

Not applicable.

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
