# Peer review of "Time-Specific Factors Influencing the Development of Asthma in Children"

_biomedicines, 2022, doi:10.3390/biomedicines10040758_

Round 1
Reviewer 1 Report
The authors wrote a review on potential factors affecting the development of asthma in children. While it was coherently written, there are several comments of the current manuscript.
1) The section on vitamins could have been elaborated more on omega3 (fish oil) supplementation, as well as the rationale on the few selected vitamins listed (fetal - C and D; infant - D only) while not including fish oil, vitamin E and etc.
2) A table summarizing the various factors at different stages of life gives a nice overview of this review.
3) On a similar note, can the authors explain why there are more factors during the fetal phase as compared to during infant/adolescent?
Author Response
We thank the referee for his/her comments.
1) We discussed more deeply the effects of omega 3 and vitamin E supplementation as the referee indicated. We are aware that our study does not detail all potential risk or protective factors. However, our choice was guided by editorial needs and mostly by the aim to investigate those factors with more solid evidence.
2) We thank for the comment on our figure.
3) We analyzed more factors during the fetal phase as compared to during infant/adolescent because the fetal period represents a critical phase in the development not only of the respiratory system but also of the immune response and the heterogeneity of the bacterial flora of the microbiome. In addition, most of the studies in literature investigated more frequently fetal factors given that the concept concerning the development of the lungs up until 18 years of age is becoming now more and more evident. Only few recent studies so far have focused on factors acting after two first years of life. Therefore, it appears evident that current scientific research has invested resources in identifying early modifiable factors. This would allow to implement preventive strategies in order to change the decline of lung function in a reversible phase.
Reviewer 2 Report
IMPORTANCE OF THE QUESTION OR SUBJECT STUDIED
The study of risk factors for asthma is a relevant topic for research.
The objectives are clearly stated.
ADEQUACY OF APPROACH
Ther are some weak points in the approach to the study.
A methods section seems necessary explaining the key points to perform the research, such as how was carried out the literature search, what key words were used, in which time frame was performed the research, what type of studies (cross-sectional, cohorts, case control, etc) were included.
There are some articles dealing wiht asthma predictive index, which deserve some comments (Castro-Rodriguez, Am J Respir Crit Care Med. 2000 Oct;162(4 Pt 1):1403-6.).
The quality of the studies included should be evaluated
It is acceptable from an ethical point of view
RESULTS
The results are clearly presented
Some important risk factors were not included (see lacking references below)
DISCUSSION
Some discussion dealing with the contradictory results for some risk factors are necessary
REFERENCES
The references are relevant. However, I think that some relevant references are lacking:
Prevalence and Risk Factors of Asthma in Preschool Children in Shanghai, China: A Cross-Sectional Study. Ren J, Xu J, Zhang P, Bao Y.Front Pediatr. 2022 Feb 9;9:793452
Diesel exhausts particles: Their role in increasing the incidence of asthma. Reviewing the evidence of a causal link. Muñoz X, Barreiro E, Bustamante V, Lopez-Campos JL, González-Barcala FJ, Cruz MJ.Sci Total Environ. 2019 Feb 20;652:1129-1138.
Investigating obesity-related risk factors for childhood asthma.
Chen YC, Su MW, Brumpton BM, Lee YL.Pediatr Allergy Immunol. 2022 Jan;33(1):e13710.
Mediterranean diet and asthma in Spanish schoolchildren. Gonzalez Barcala FJ, Pertega S, Bamonde L, Garnelo L, Perez Castro T, Sampedro M, Sanchez Lastres J, San Jose Gonzalez MA, Lopez Silvarrey A.Pediatr Allergy Immunol. 2010 Nov;21(7):1021-7.
The Dietary Inflammatory Index and asthma burden in children: A latent class analysis.
Cilluffo G, Han YY, Ferrante G, Dello Russo M, Lauria F, Fasola S, Montalbano L, Malizia V, Forno E, La Grutta S.Pediatr Allergy Immunol. 2022 Jan;33(1):e13667.
A clinical index to define risk of asthma in young children with recurrent wheezing. Castro-Rodríguez JA, Holberg CJ, Wright AL, Martinez FD. Am J Respir Crit Care Med. 2000 Oct;162(4 Pt 1):1403-6.
Rhinovirus Infection and Familial Atopy Predict Persistent Asthma and Sensitisation 7 Years after a First Episode of Acute Bronchiolitis in Infancy.
Magnier J, Julian V, Mulliez A, Usclade A, Rochette E, Evrard B, Amat F, Egron C.
Children (Basel). 2021 Sep 26;8(10):850.
Lemanske RF. The childhood origins of asthma (coast) study. Pediatr Allergy Immunol. (2002) 13:38–43
Biagini Myers JM, Schauberger E, He H, Martin LJ, Kroner J, Hill GM, et al. A pediatric asthma risk score to better predict asthma development in young children. J Allergy Clin Immunol. (2018) 143:1803–10
Exposure to paracetamol and asthma symptoms. Gonzalez-Barcala FJ, Pertega S, Perez Castro T, Sampedro M, Sanchez Lastres J, San Jose Gonzalez MA, Bamonde L, Garnelo L, Valdes L, Carreira JM, Moure J, Lopez Silvarrey A.Eur J Public Health. 2013 Aug;23(4):706-10.
Impact of parental smoking on childhood asthma. Gonzalez-Barcala FJ, Pertega S, Sampedro M, Lastres JS, Gonzalez MA, Bamonde L, Garnelo L, Castro TP, Valdés-Cuadrado L, Carreira JM, Moure JD, Silvarrey AL.J Pediatr (Rio J). 2013 May-Jun;89(3):294-9.
Truck traffic related air pollution associated with asthma symptoms in young boys: a cross-sectional study. Gonzalez-Barcala FJ, Pertega S, Garnelo L, Castro TP, Sampedro M, Lastres JS, San Jose Gonzalez MA, Bamonde L, Valdes L, Carreira JM, Silvarrey AL.Public Health. 2013 Mar;127(3):275-81.
Caesarean delivery and risk of childhood asthma development: meta-analysis.
Wypych-Ślusarska A, Niewiadomska E, Oleksiuk K, Krupa-Kotara K, Głogowska-Ligus J, Słowiński J.Postepy Dermatol Alergol. 2021 Oct;38(5):819-826.
Factors Associated With Childhood Asthma and Wheeze in Chinese Preschool-Aged Children.
Deng X, Yang M, Wang S, Wang Q, Pang B, Wang K, Zhang Z, Niu W.Front Med (Lausanne). 2021 Nov 4;8:742581.
Childhood asthma- pathogenesis and phenotypes. Pijnenburg MW, Frey U, De Jongste JC, Saglani S.Eur Respir J. 2021 Oct 28:2100731.
Parental socioeconomic status and asthma in children: Using a population-based cohort and family design. Caffrey Osvald E, Gong T, Lundholm C, Larsson H, Bk B, Almqvist C.Clin Exp Allergy. 2022 Jan;52(1):94-103.
Association of Treated and Untreated Gastroesophageal Reflux Disease in the First Year of Life with the Subsequent Development of Asthma. Cantarutti A, Barbiellini Amidei C, Valsecchi C, Scamarcia A, Corrao G, Gregori D, Giaquinto C, Ludvigsson JF, Canova C.Int J Environ Res Public Health. 2021 Sep 13;18(18):9633.
Association of the Risk of Childhood Asthma at Age 6 With Maternal Allergic or Immune-Mediated Inflammatory Diseases: A Nationwide Population-Based Study.
Yang DH, Chin CS, Chao WC, Lin CH, Chen YW, Chen YH, Chen HH.Front Med (Lausanne). 2021 Aug 2;8:713262
GRAMMAR AND STYLE
The writing is easy to follow.
ABSTRACT
I think that the main results should be included in the abstract
Author Response
We thank the referee for his/her comments and suggestions.
Considering your precious comment, we added a concise section on the methods used in our research. We tried to describe the keywords used for the database search, the type of studies included, and the time interval considered. However, we would like to underline that our work is a narrative review, which identifies several studies (not all of the studies present in literature) describing our topic of interest differently from a systematic review.
We included in our list, other relevant risk factors, such as diesel exhausts particles, Mediterranean diet and dietary inflammatory index, parental socioeconomic status, gastroesophageal reflux disease. In addition, we discussed more deeply some factors already mentioned and inserted other contrasting studies.
We added in the abstract the main results of our narrative review as the referee indicated.
Reviewer 3 Report
Russo et al. analyzed the factors affecting asthma incidence in children and adolescents from various angles through an extensive literature review and suggested preventive and therapeutic initiatives.
1. It would be more helpful to present the title in more detail. For example, 'Time-specific factors influencing the development of asthma in children and adolescents.'
2. It will likely be more readable to express related parts of the 'Fetal and perinatal exposure' section sub-items in groups. For example, Vitamin E and Vitamins can be grouped into one sub-section.
3. Some abbreviations do not contain full names when first described. Please check and supplement it.
4. When expressing long numbers with the division of digits, please mark it with a comma instead of a period.
5. Please check and correct spelling errors and typos throughout the manuscript.
Author Response
We thank the referee for his/her very precious suggestions.
1) We changed the title as the referee suggested.
2) We inserted in a unique paragraph the description of the similar factors (i.e. Vitamin E with vitamins). We only preferred to keep separate the paragraph describing prenatal farming exposure as it is a protective factor differently from other environmental risk factors.
3) we added the full name in abbreviations.
4) we replaced the period with the comma when expressing long numbers.
5) we corrected spelling errors and typos.
Round 2
Reviewer 2 Report
All my concerns were correctly addressed